# Rehabilitation outcomes following tail-fluke amputation in an Indo-Pacific bottlenose dolphin: A welfare-centered approach

Suguru Higa[1], Sayaka Takahashi[1], Eri Nakashima[1], Yui Kurosu[1], Haruka Ikeshima[1], Ryota Yagi[1], Nihiro Adachi[1], Keiichi Ueda[1], Hirobumi Umeyama[2], Hitoshi Yamamoto[2], Yukinori Nakakita[2], Kazuma Tochigi[2], Taihei Kagawa[2], Shin-ichiro Oka[1]*

1 Aquarium Management Center, Okinawa Churashima Foundation, Motobu, Okinawa, Japan,
2 Bridgestone Corporation, Chuo-ku, Tokyo, Japan

* sh-oka@okichura.jp

## Abstract

Tail fluke loss in cetaceans compromises locomotion and impairs social functioning, posing serious welfare challenges. Rehabilitation strategies that address both physical performance and behavioral reintegration are essential for improving quality of life in affected individuals. This study evaluated the outcomes of a structured, two-phase rehabilitation program applied to Sami, an adult Indo-Pacific bottlenose dolphin (*Tursiops aduncus*) that underwent tail-fluke amputation, with an emphasis on locomotor function, behavioral adaptation, and long-term welfare. The rehabilitation protocol comprised two phases. Phase 1 focused on restoring vertical tail-beat locomotion through a combination of range-of-motion exercises and the use of a custom-designed prosthetic tail fluke. Swimming performance was quantitatively assessed using biologging devices under three conditions: without tail flukes, with the prosthetic tail, and in healthy conspecifics. Phase 2 introduced structured cohabitation with familiar individuals to promote social reintegration. Behavioral data were collected before and after rehabilitation to evaluate affiliative engagement and activity patterns. While maximum swim speed remained lower than in healthy individuals, propulsion per stroke significantly improved with prosthetic use. Notably, the subject dolphin retained species-typical vertical tail-beat motion even after prosthesis discontinuation, indicating motor pattern adaptation. Post-rehabilitation, affiliative behavior increased to 17% of total observed activity—more than twice that of a healthy control—while resting behavior markedly declined. No aggression or abnormal behaviors were observed. This study demonstrates that a welfare-centered, multi-phase rehabilitation framework can effectively promote both functional recovery and social reengagement in dolphins with severe caudal injuries. The long-term retention of adaptive locomotor and social behaviors highlights the potential of integrative approaches to enhance the quality of life in physically compromised cetaceans.

**Data availability statement:** All relevant data are within the manuscript and its Supporting Information files.

**Funding:** This study was supported by independent research budgets of the Okinawa Churashima Foundation (to SH, ST, EN, YK, HI, RY, NA, KU, and SO) and Bridgestone Corporation (to HU, HY, YN, KT, TK). The funders had no role in study design, data collection and analysis, decision to publish, or preparation of the manuscript.

**Competing interests:** This study was supported by independent research budgets of the Okinawa Churashima Foundation and Bridgestone Corporation. Bridgestone Corporation contributed to the development of the prosthetic tail. This does not alter our adherence to PLOS ONE policies on sharing data and materials. There are no patents, products in development or marketed products associated with this research to declare.

## Introduction

In recent years, the concept of animal welfare has gained significant attention in marine biology and captive animal care. Ensuring the physical and psychological well-being of animals under human care is now a central priority [1–3]. The current framework of animal welfare evaluation increasingly incorporates the multidimensional perspective described in the Five Domains model [4]. Among marine mammals, dolphins are known for their advanced intelligence, strong social bonds, and specific physical needs critical for maintaining their quality of life (QOL) [2,5,6]. However, severe injuries such as the loss of tail flukes pose significant challenges to their mobility, social integration, and overall welfare, underscoring the need for innovative rehabilitation strategies.

While anecdotal reports have documented the application of prosthetic devices in dolphins—such as in cases at the Okinawa Churaumi Aquarium and Clearwater Marine Aquarium—these interventions have not been systematically evaluated [2, 7]. In particular, long-term outcomes related to psychological well-being, social reintegration, and behavioral adaptability remain largely unexplored.

Globally, recent reviews have emphasized that marine mammal rehabilitation efforts still lack standardized clinical frameworks and consistent reporting of welfare outcomes [8]. These studies also highlight the need for welfare-centered, evidence-based rehabilitation protocols and transparent post-treatment evaluations. Building on this context, the present study addresses these challenges by providing a structured, welfare-focused rehabilitation case involving an Indo-Pacific bottlenose dolphin (Tursiops aduncus) that underwent tail-fluke amputation followed by prosthetic adaptation.

To our knowledge, this represents the first systematic assessment of such an approach in this species. The research aims to evaluate the impact of a prosthetic tail on restoring swimming ability, enhancing social reintegration, and improving overall welfare. By combining detailed behavioral observations with objective assessments of psychological and social outcomes, this study advances the understanding of holistic welfare approaches in marine mammal rehabilitation.

This work also holds broader significance. It demonstrates how integrating welfare-centered practices with technological innovation can address the complex needs of injured marine animals. Additionally, it contributes to the development of standardized methods for assessing animal welfare, providing a model for facilities worldwide. By bridging gaps in knowledge and practice, this study not only improves the quality of life for dolphins like the one in this case but also paves the way for more effective and compassionate care for marine mammals globally.

Although this study focuses on a single individual, it is designed not as an anecdotal case report but as a hypothesis-driven, systematically structured investigation incorporating quantitative and behavioral assessments. Given the rarity and ethical limitations associated with studying severe caudal injuries in cetaceans, such opportunities for longitudinal evaluation under controlled conditions are extremely limited. Therefore, this study serves as a scientifically grounded pilot for evaluating rehabilitation strategies that are potentially generalizable to other individuals or species under

professional care. Our goal is to contribute a reproducible framework that can inform future multi-individual studies and enhance rehabilitation practices across institutions.

## Materials and methods

### Ethical statement

This study was conducted in accordance with the Ethical Guidelines for the Conduct of Research on Animals by Zoos and Aquariums issued by the World Association of Zoos and Aquariums (WAZA) and the Code of Ethics established by the Japanese Association of Zoos and Aquariums (JAZA). It also adhered to the Animal Care Guidelines of the Okinawa Churashima Foundation, which oversees the Okinawa Churaumi Aquarium. Additionally, the study was approved by the Animal Experiment Committee established by the Okinawa Churashima Foundation (Permission Number: AT22003). These guidelines and committee oversight emphasize the importance of a thorough understanding and careful management of animal welfare. The acts involved in this study did not negatively impact the welfare of the dolphins involved.

### Subject Dolphin

The subject of this study was Sami, a captive-born adult female Indo-Pacific bottlenose dolphin (*Tursiops aduncus*) housed at the Okinawa Churaumi Aquarium. In February 2021, she underwent surgical amputation of the tail flukes due to trauma-induced necrosis. The rehabilitation process enabled a structured evaluation of prosthetic-assisted recovery in cetaceans with severe caudal injury, focusing on swimming function, social reintegration, and overall welfare.

In September 2020, the subject dolphin sustained a significant blunt trauma to her tail flukes after colliding with the pool wall. The resulting injury developed into a complex infection, leading to progressive necrosis of the affected tissue. On February 2021, post-injury, the tail flukes were surgically amputated, leaving only the central portion intact. The impact also caused a rupture of the ventral tendon on the right side of the caudal vertebrae, resulting in an asymmetrical deformation of the tail. Specifically, the distal portion of the vertebrae became twisted upward and to the right (Fig 1).

Following the completion of the surgical amputation in June 2021, the subject dolphin initially attempted to swim using a lateral (side-to-side) tail motion. However, this failed to generate sufficient propulsion, and her voluntary swimming activity gradually declined. At the same time, her engagement in social interactions with other dolphins also decreased markedly. This state persisted for approximately one month. At the time of the incident, the subject dolphin measured 237 cm in length and weighed 143 kg.

### Rehabilitation 1: Restoration of swimming ability using a prosthetic Tail flukes

In line with the case of Fuji, a Pacific bottlenose dolphin at our aquarium whose quality of life (QOL) was significantly improved through the use of prosthetic tail flukes [2], this study prioritized the restoration of swimming ability through the application of a prosthetic device. The subject dolphin had an asymmetrical distal end of the tail stump, requiring a secure attachment system capable of withstanding the forces generated during propulsion. A socket-type attachment was designed to encase the entire caudal peduncle, onto which rubber prosthetic tail flukes—shaped to resemble natural flukes—were mounted (Fig 1, S1 Video).

Following the loss of her tail flukes, the subject dolphin developed a compensatory swimming pattern characterized by pronounced lateral body undulation (S2 Video). While this strategy may have enabled limited locomotion, it imposed abnormal strain on muscle groups and tendons not typically engaged during normal cetacean swimming. To mitigate the risk of injury and to support functional recovery, rehabilitation efforts focused on reestablishing a species-typical stroke pattern involving vertical tail movements, which are essential for efficient propulsion and overall biomechanical balance.

To address this, a rehabilitation program was implemented between July 2021 and May 2022 to increase vertical tail flexibility and restore an appropriate stroke axis. During these sessions, the pool was drained and the dolphin's body was

 

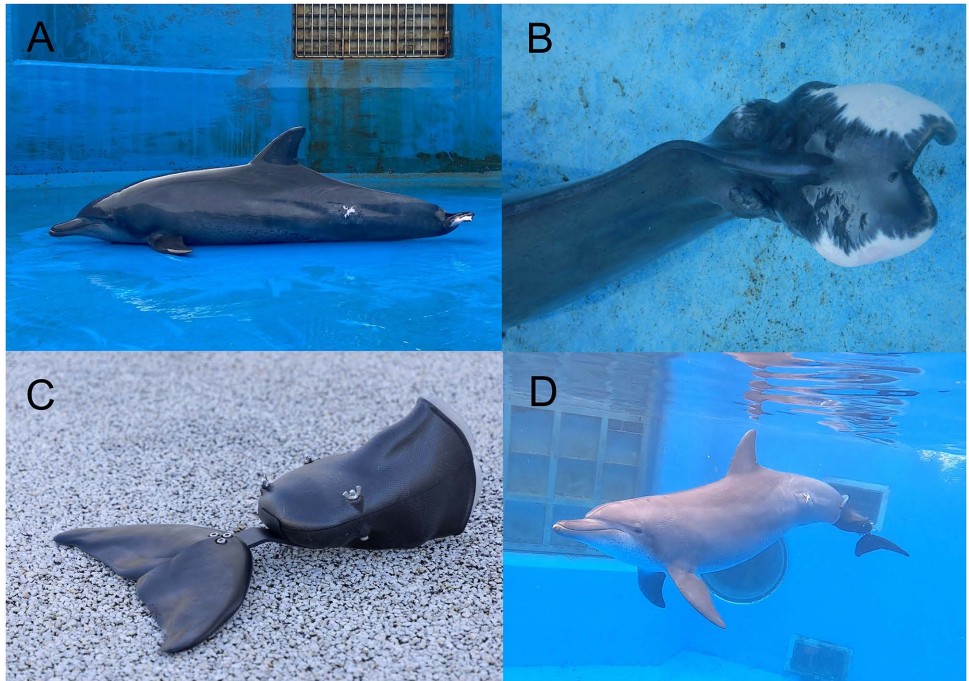

**Fig 1. Tail flukes amputation and prosthetic application in the Indo-Pacific bottlenose dolphin. (A)** Lateral view of the subject dolphin following surgical removal of the tail flukes due to trauma-induced necrosis. **(B)** Close-up of the residual tail stump showing asymmetrical deformation. **(C)** Custom-designed prosthetic tail flukes developed for attachment to the caudal peduncle. **(D)** The dolphin swimming with the prosthetic tail flukes attached, exhibiting vertical tail-beat motion.

gently restrained. A single trainer manually guided the distal tail region through a full vertical range of motion—up and down—until natural resistance was encountered. Each session lasted approximately 20 minutes and was conducted one to five times per week, for a total of 30 sessions (S3 Video). No aversive or avoidance behaviors were observed during this phase.

Following this stretching rehabilitation, habituation training to the prosthetic device was carried out between May and December 2022. The prosthetic tail was attached, and the subject dolphin was allowed to swim freely in her exhibit pool for 5–15 minutes per session. This training was conducted 1–5 times per week for a total of 43 sessions. The goal of this phase was to promote physical adaptation and establish a positive behavioral association with the prosthesis.

## Evaluation of swimming efficiency using prosthetic Tail flukes

To evaluate the effect of a prosthetic tail on swimming performance, we compared three conditions: (1) a dolphin without tail flukes (the subject dolphin), (2) the same dolphin with prosthetic tail flukes, and (3) a healthy individual of the same species. In all trials, the dolphins were encouraged to swim at maximum effort along the perimeter of the pool. Swimming efficiency was assessed using data recorded by animal-borne data loggers.

The loggers used were OR1400−3MPD3GT units manufactured by Little Leonardo Corporation. Two loggers were attached to each dolphin using suction cups—one placed just below the dorsal fin to record swim speed, and the other near the tail region to measure stroke frequency based on the periodicity of acceleration along the z-axis.Data loggers recorded swim speed, stroke frequency, and tri-axial acceleration (x, y, and z axes). In the present analysis, the periodicity of acceleration along the longitudinal (z) axis was used to calculate stroke frequency, while swim speed was obtained from the synchronized data.

For the subject dolphin, nine trials were conducted without the prosthetic tail flukes and eight trials with it. In addition, nine trials were conducted in total with two healthy Indo-Pacific bottlenose dolphins. For each trial, we extracted one complete tail stroke corresponding to the moment of maximum swim speed for analysis.

Swimming efficiency was quantified using the relative propulsion distance per stroke (RPDS), a dimensionless index defined as:

$$RPDS = \frac{\text{Swim speed } (\text{m s}^{-1})}{\text{Body length (m)} \times \text{Stroke frequency (stroke s}^{-1})}$$

This index represents the relative distance traveled per stroke, normalized by body length and stroke frequency. Bainbridge [9] demonstrated a strong relationship between swim speed, body length, and tail beat frequency, supporting the validity of these parameters. The RPDS was therefore considered a particularly suitable indicator for evaluating and comparing the swimming efficiency of a dolphin without tail flukes, one equipped with prosthetic tail flukes, and healthy conspecifics.

However, the evaluation of swimming efficiency using the prosthetic tail was discontinued in January 2023 to prevent the progression of a localized infection observed at the distal end of the tail stump. This decision was made as a precautionary medical measure, and the prosthetic tail has not been used since, to allow for proper treatment and healing. Statistical comparisons of locomotor parameters among the rehabilitation phases were conducted using one-way analysis of variance (ANOVA) followed by Tukey's honest significant difference (HSD) post hoc test. All analyses were performed in Python (version 3.10) using standard scientific libraries.

## Rehabilitation 2: Restoration of Social Behavior

Social relationships are known to play a critical role in the welfare of Indo-Pacific bottlenose dolphins [5,6]. Following the surgical removal of the tail flukes, the subject dolphin was housed alone throughout Rehabilitation Phase 1, which focused on restoring swimming ability. During this period, a short-term cohabitation trial was conducted with a familiar conspecific that had previously lived with the subject dolphin. The aim of this trial was to promote the reacquisition of natural social behaviors. However, the attempt was discontinued due to the emergence of aggressive behaviors, such as chasing and physical contact directed toward the subject dolphin, as she was unable to synchronize her movements with the healthy individual.

As Rehabilitation Phase 1 progressed, the subject dolphin began to exhibit consistent, voluntary swimming characterized by stable vertical tail movements, indicating a recovery of motivation. Based on this improvement, a second short-term cohabitation trial was implemented. This time, no aggressive or avoidance behaviors (e.g., chasing or withdrawal) were observed. In light of this outcome, a gradual reintroduction to social interactions was initiated as Rehabilitation Phase 2, with the goal of restoring social engagement.

Rehabilitation Phase 2 was conducted over a period of 53 days between September and October 2024. The subject dolphin was housed in a large outdoor pool (maximum diameter: 25 m; depth: 6 m; water volume: 1,750 m³) together with a diverse group of conspecifics and related species, including *Tursiops aduncus*, *Tursiops truncatus*, Pseudorca crassidens, and *Steno bredanensis*—all female individuals with whom she had previously established affiliative relationships. To promote the restoration of sociality through natural interactions, no special treatment was given to the subject dolphin during the cohabitation period, and human intervention in inter-animal dynamics was deliberately avoided. The number of cohabiting individuals (excluding the subject dolphin) fluctuated throughout the rehabilitation period due to management requirements: four animals were present for 26 days, five for 22 days, three for 3 days, and six for 2 days. Qualitative behavioral observations were conducted five times daily, generally aligned with the five scheduled feeding sessions. These observations focused on spontaneous interactions, behavioral changes, and social dynamics among dolphins and served as a basis for evaluating the progression of her social reintegration in a naturalistic setting.

To quantitatively assess behavioral changes before and after Rehabilitation Phase 2, focal observations were conducted on the subject dolphin over two distinct periods: four days between November 2021 and January 2022 (pre-rehabilitation), and three days between October and November 2024 (post-rehabilitation). During the pre-rehabilitation period, the subject dolphin was housed with a female rough-toothed dolphin (*Steno bredanensis*) and bottlenose dolphins (*Tursiops truncatus*), while in the post-rehabilitation period, she was cohabiting with female bottlenose dolphins and false killer whales (*Pseudorca crassidens*). All cohabiting animals had previously exhibited affiliative relationships with the subject dolphin prior to her injury. In both periods, cohabitation took place in an outdoor tank measuring 14 meters in length, 3 meters in depth, and with a water volume of 400 m³.

The focal animal's behavioral state was recorded every 5 minutes over a 1-hour period using scan sampling. Observations were consistently conducted from 12:30–13:30, a non-feeding period, to avoid the influence of feeding motivation. Because the behavioral state was recorded objectively at fixed 5-minute intervals using scan sampling, observer bias and recording errors were minimized. All sessions were conducted while the dolphin was not wearing prosthetic tail flukes.

To evaluate the dolphin's psychological and social state, six behavioral categories were used, based on the definitions provided by Miller et al. [6]: resting, affiliative behavior, aggressive behavior, repetitive behavior, high-energy behavior, and others (e.g., routine swimming, object play, exploratory behavior). As a behavioral control, the same behavioral observations were also conducted on a healthy adult female Indo-Pacific bottlenose dolphin over seven days between October 2024 and January 2025. This individual had been under human care at the Okinawa Churaumi Aquarium since 1975 and was the biological mother of the subject dolphin. While her exact age is unknown, her long-term residency and environmental familiarity were considered appropriate for use as a behavioral reference in this study.

## Result

### Rehabilitation phase 1

Rehabilitation Phase 1, which focused on expanding the vertical range of tail motion and correcting the stroke axis from horizontal to vertical, required approximately 11 months to complete. As a result of this intervention, the subject dolphin regained the ability to swim with a consistent vertical tail beat. Habituation to the prosthetic tail flukes also progressed smoothly. Within approximately 3 months from the start of this phase, the subject dolphin demonstrated stable and coordinated swimming while wearing the prosthetic (S4 Video). Notably, even when not wearing the prosthetic tail flukes, the subject dolphin no longer exhibited the snake-like lateral body undulation that was previously observed immediately following tail loss. This behavioral change suggests that the vertical tail stroke pattern was retained even in the absence of the prosthetic device, indicating the effectiveness of the rehabilitation in promoting lasting motor adaptation.

#### Swimming efficiency using prosthetic tail flukes

Fig 2 illustrates the results of high-speed swimming trials under three conditions: without tail flukes (No Fin), with prosthetic tail flukes, and in a healthy control individual. The maximum swimming speed was significantly lower in both the no fin and prosthetic conditions compared to the control group ($p < 0.001$), with no significant difference between the no fin and Prosthetic groups.

Stroke frequency, measured as the number of tail beats per second, did not differ significantly between the no fin and control groups ($p = 0.53$), but was significantly reduced in the prosthetic condition compared to both no fin and control ($p < 0.001$).

Relative propulsion distance per stroke (RPDS), which quantifies body-length-normalized distance traveled per tail stroke, differed significantly among all three groups ($p < 0.001$). The no fin condition showed the lowest RPDS values, while the prosthetic condition was significantly higher and closer to Control levels. In some trials, RPDS values in the prosthetic condition were comparable to or even exceeded those observed in healthy individuals.

These results indicate that while the prosthetic tail did not contribute substantially to increasing swim speed—likely due to increased drag and reduced flexibility—it partially restored propulsion efficiency lost due to tail loss. Notably, even

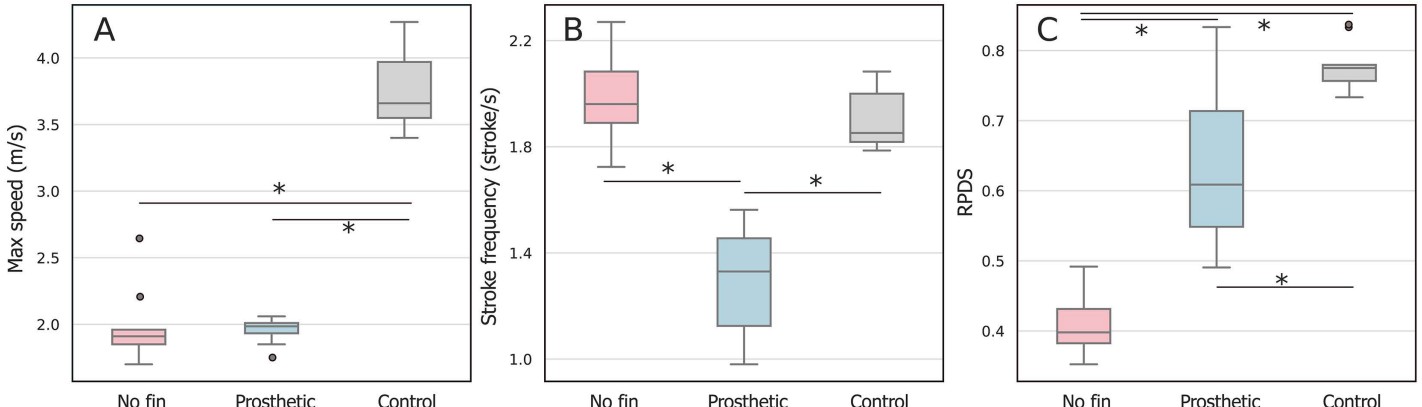

**Fig 2. Boxplots of maximum swim speed (A), stroke frequency (B), and relative propulsive distance per stroke (RPDS) (C) measured by accelerometer under three conditions: No Fin, Prosthetic Tail, and Control (a healthy individual with intact tail flukes).** Asterisks indicate statistically significant differences between groups based on Tukey's HSD test ($p < 0.05$). Boxes represent the interquartile range (IQR), horizontal lines inside boxes indicate medians, and whiskers extend to $1.5 \times$ IQR. Outliers are plotted as individual points.

though stroke frequency was reduced in the prosthetic condition, the distance traveled per stroke was improving the effectiveness of the prosthetic tail as a functional assistive device for locomotion.

Since January 2023, the use of the prosthetic tail has been discontinued for medical reasons. Nevertheless, until June 2025, when the individual died due to unrelated chronic conditions, primarily pulmonary in nature, the subject dolphin consistently maintained the vertical tail-beat swimming pattern acquired through rehabilitation, suggesting that the motor adaptation had been retained in the long-term.

### Rehabilitation Phase 2

During Rehabilitation Phase 2, which aimed to restore social behaviors through active cohabitation, affiliative interactions began to emerge shortly after the initiation of the program. Behaviors such as synchronous swimming and close-contact positioning—completely absent prior to rehabilitation—were consistently observed. In addition, instances of drafting behavior—defined as one individual swimming in the hydrodynamic wake of another—were documented during cohabitation (S5 Video). As this behavior entails coordinated movement and sustained proximity, it is considered a form of affiliative interaction indicative of social cohesion. Quantitative behavioral analysis (Fig 3) revealed that prior to rehabilitation, the subject dolphin spent most of the observation time resting motionlessly at the surface, with no affiliative or social interactions recorded. In contrast, near the end of Rehabilitation Phase 2, the proportion of resting behavior had markedly decreased, while the "Others" category nearly doubled. Notably, affiliative behaviors accounted for approximately 17% of total observed behaviors. Within the "Others" category, the majority consisted of solitary swimming, with a smaller portion comprising exploratory and solitary play behaviors.

Compared to a healthy adult *Tursiops aduncus* observed under similar conditions, the subject dolphin demonstrated more than double the frequency of affiliative behavior. No aggressive or repetitive behaviors were recorded throughout the observation period, suggesting that the subject dolphin had re-established social engagement in a stable and positive manner.

Remarkably, these affiliative behaviors—including synchronous swimming and close-contact positioning—were observed even during the final stage of her life, while she was affected by a pulmonary condition, continuing up to the day before her death.

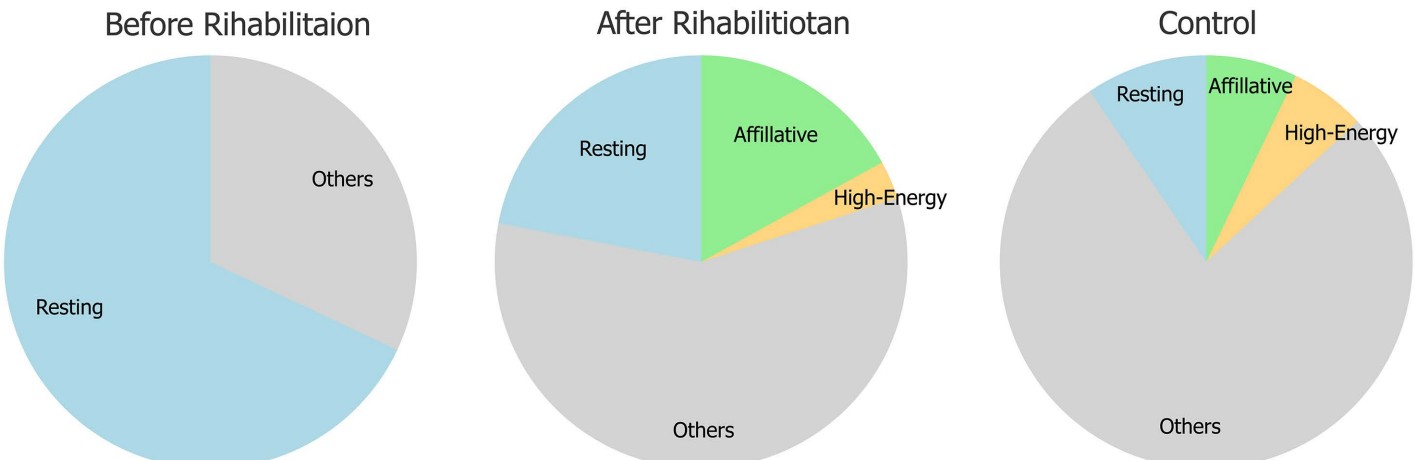

**Fig 3. Behavioral composition of a rehabilitated Indo-Pacific bottlenose dolphin before and after rehabilitation, compared with a healthy control.** Pie charts represent the proportion of six behavioral categories—resting, affiliative, high-energy, and others—observed during focal sampling sessions. "Others" includes routine swimming, solitary play, and exploratory behaviors.

## Discussion

### Ethical considerations in rehabilitation

Ethical considerations were integral to decision-making throughout the rehabilitation process. The duration of prosthesis use was determined through continuous medical assessment and welfare monitoring, and the device was ultimately discontinued in accordance with the precautionary principle to prevent further infection at the stump site. Similarly, the timing of social reintroduction was guided by the dolphin's behavioral stability, health status, and stress indicators to ensure optimal welfare conditions. These decisions were made collaboratively by veterinarians and animal keepers, reflecting a welfare-centered and precautionary approach to rehabilitation.

### Evaluation of physical recovery and social behavior

Compared with previously reported rehabilitation efforts that primarily emphasized prosthetic design or short-term locomotor performance, the present study highlights the advantages of a structured two-phase rehabilitation framework. By first restoring species-typical locomotor patterns through targeted physical rehabilitation (Phase 1), and subsequently promoting social reintegration through controlled cohabitation (Phase 2), this approach facilitated not only functional adaptation but also sustained social and behavioral recovery. This sequential design may be particularly important for highly social species, in which locomotor competence and social acceptance are closely linked.

The rehabilitation program implemented for the subject dolphin resulted in significant improvements in both her physical performance and social engagement. In terms of swimming ability, the structured training allowed her to recover a vertical tail-beat locomotion, which had been lost following the injury. Although swimming speed in both the No Fin and Prosthetic conditions remained significantly lower than in healthy dolphins (Fig 2), the use of the prosthetic tail partially restored propulsion efficiency, as evidenced by a significantly higher RPDS compared to the No Fin condition. Notably, the subject dolphin continued to exhibit a stable vertical tail-beat pattern even after the prosthesis had been discontinued due to medical considerations, suggesting lasting motor adaptation.

Although maximum swim speed did not recover to levels observed in healthy individuals, this outcome reflects the functional scope of the prosthetic intervention rather than a failure of rehabilitation. The primary goal of the prosthetic

tail was to restore species-typical vertical tail-beat locomotion and improve swimming efficiency, rather than to achieve peak performance. The significant improvement in propulsion efficiency per stroke and the long-term retention of vertical tail-beat motion indicate successful motor adaptation. From a welfare perspective, such sustainable and efficient locomotion is likely more relevant to long-term fitness and social participation than maximum swimming speed.

In the domain of social behavior, progress was even more striking. At the end of Rehabilitation Phase 2, affiliative behaviors—such as synchronous swimming and close-contact positioning—accounted for approximately 17% of total observed behavior (Fig 3). Prior to rehabilitation, such interactions were entirely absent, and the subject dolphin spent nearly all her time resting motionlessly at the surface. In contrast, post-rehabilitation observations showed a substantial reduction in resting and an increase in active behaviors, including solitary swimming, exploratory movement, and solitary play. Within the "Others" category, the majority consisted of solitary swimming, with a smaller portion comprising exploratory and solitary play behaviors.

Moreover, her social activity time fell within the range observed in wild dolphins, which spend approximately 4–31% of their time engaged in social behaviors [6]. Compared with a healthy adult *Tursiops aduncus* observed under similar conditions, the subject dolphin demonstrated more than double the frequency of affiliative behavior. Her successful reintegration into the social group underscores the rehabilitation program's effectiveness in addressing both physical and social challenges, and it contributed significantly to her overall recovery and quality of life (QOL). Notably, both the vertically undulating swimming pattern and affiliative behaviors were retained until the final stage of her life, even after the progression of a fatal pulmonary condition. The long-term retention of vertical locomotion and affiliative behaviors indicates the potential for sustained adaptation following targeted rehabilitation interventions.

## Psychological changes and resilience

Beyond these physical and behavioral outcomes, the rehabilitation process appeared to foster notable psychological changes. The repeated success in adapting to the prosthetic tail flukes, followed by the maintenance of learned motor patterns even after its discontinuation, may be indicative of enhanced self-efficacy—the belief in one's ability to perform essential tasks [10]. This concept, originally developed in human psychology, has gained increasing relevance in the context of animal rehabilitation, particularly for cognitively advanced species such as dolphins [11].

Another critical factor in her recovery was the successful generalization of learned behaviors and the restoration of a species-typical movement pattern. Skills acquired during prosthetic-assisted swimming—such as vertical tail-beat motion and coordinated body control—were effectively transferred to situations where the prosthesis was no longer used. This continuity suggests the development of robust motor learning and supports Shepard's theory of generalization [12], whereby behaviors learned in one condition can be applied to analogous situations through shared psychological processes. Importantly, the regained vertical tail movement may have played a key role not only in sustaining physical mobility but also in facilitating social reintegration. At present, however, direct empirical evidence on whether dolphins exhibit preferential care or aggressive responses toward handicapped conspecifics remains limited. It is plausible that movement patterns consistent with species-typical norms contribute to social recognition and acceptance within animal groups. In some mammalian species, disruptions to such behavioral cues have been linked to social rejection or exclusion. For instance, chimpanzees that exhibit aberrant or socially inappropriate behavior are known to be actively shunned by group members [13], and African elephants that experience early-life social disruption have shown non-normative behavior and difficulties in reintegrating into social groups due to deficits in learned social and motor patterns [14]. Furthermore, abnormal locomotor patterns—especially those that deviate from species-typical mechanics—may trigger avoidance or aggression from peers [15] By regaining vertical tail-beat locomotion, the subject dolphin may have reduced the perceptual incongruence that could otherwise have led to negative social responses, ultimately promoting social acceptance and the emergence of affiliative behaviors during cohabitation.

**The role of positive reinforcement**

Throughout both rehabilitation phases, positive reinforcement played a central role. Reward-based training not only facilitated prosthetic adaptation but also reinforced a positive association with swimming and physical engagement. The continuation of voluntary swimming and exploration after the discontinuation of the prosthesis further suggests that intrinsic motivation had been established. Parallels can be found in human and non-human primate rehabilitation, where structured positive reinforcement has been shown to enhance motivation, reduce stress, and improve adaptability [16,17]. These findings underscore the cross-species relevance of reinforcement-based methodologies in promoting behavioral resilience and psychological well-being.

**Implications for Animal Welfare and Rehabilitation**

Although prosthetic design is an important consideration, detailed structural and technical aspects of the artificial tail fluke were beyond the scope of the present welfare-focused study and will be described in a separate manuscript. Instead, the present findings emphasize the importance of a holistic approach to rehabilitation that addresses both physical and psychological domains.

Reintegration into a social group was a critical component of recovery, as social affiliation and culturally transmitted behaviors are vital to the well-being of cetaceans [18]. In highly social species like the Indo-Pacific bottlenose dolphin, providing opportunities for meaningful social engagement is essential for long-term welfare. The observed resilience and adaptability further highlight the role of cognitive processing in animal welfare, as discussed by Duncan and Petherick [15]. Structured rehabilitation programs that incorporate opportunities for learning, problem-solving, and autonomous interaction can enhance not only physical recovery but also psychological health and behavioral flexibility.

Although prosthetic applications in marine mammals other than dolphins have not been reported, similar attempts have been made in sea turtles that have lost their flippers. These studies, however, have largely focused on the engineering and hydrodynamic aspects of prosthetic design rather than on behavioral or welfare outcomes [19,20]. In contrast, the present study demonstrates how prosthetic technology can be integrated into a welfare-centered rehabilitation program that simultaneously addresses locomotor function, social behavior, and psychological recovery.

The observed outcomes underscore the value of structured, welfare-centered interventions in promoting recovery among cetaceans with severe injuries. The use of multidimensional metrics—including activity patterns, social behavior, and adaptive functioning—may support more standardized and species-appropriate assessments of rehabilitation success. Such frameworks could improve not only the design of future interventions but also transparency and accountability in animal welfare practices [21].

Although the subject dolphin showed sustained behavioral stability and social engagement following rehabilitation, ensuring long-term welfare remains crucial when individuals have undergone severe physical trauma. Continuous assessment of physical condition, social relationships, and behavioral diversity can help detect subtle signs of discomfort, social isolation, or chronic stress. Welfare frameworks such as the Five Domains model [4] highlight the need for multidimensional evaluation and management responsive to each animal's mental state. Moreover, recent reviews of marine mammal rehabilitation emphasize the paucity of robust long-term post-release and welfare data [8], reinforcing the imperative for ongoing monitoring. Adaptive management—guided by systematic welfare data—allows caretakers to modify group composition, enrichment programs, or medical interventions in response to changing welfare needs. Establishing longitudinal welfare datasets across facilities would further enhance our understanding of recovery trajectories and support evidence-based refinement of rehabilitation practices for marine mammals.

**Practical guidelines for dolphin rehabilitation**

Recent global reviews have highlighted the lack of standardized clinical frameworks and quantitative welfare assessments in marine mammal rehabilitation programs [8].

This study presents a structured, welfare-centered protocol that directly addresses this gap, providing a practical example of how standardization can be implemented in real-world settings. The experience from this case offers practical insights for facilities managing dolphins with severe caudal injuries.

A two-phase rehabilitation framework—comprising physical and social recovery—proved effective in guiding clinical decisions and monitoring progress.

During Phase 1 (physical rehabilitation), wound healing and muscle recovery should be prioritized, and the timing of prosthesis introduction should be determined cautiously, considering infection risk and skin tolerance.

During Phase 2 (social rehabilitation), gradual reintroduction to conspecifics and careful behavioral monitoring are essential to ensure social stability and psychological well-being.

Establishing objective behavioral and locomotor indicators to assess progress at each phase would further strengthen the reproducibility of such approaches. These recommendations could serve as a practical foundation for developing future standardized clinical protocols in marine mammal rehabilitation.

Implementation of such rehabilitation programs requires coordinated collaboration among veterinarians, animal trainers, and engineers to ensure safety and welfare at each stage. Staff should receive specialized training in prosthetic handling, behavioral monitoring, and welfare assessment to minimize stress during interventions. Facilities must be equipped with adaptable pools that allow controlled water depth for physical therapy, as well as access to materials and fabrication expertise for custom prosthetic design. Although initial development costs—such as prosthetic fabrication and data-logging equipment—can be considerable, these are often offset by the long-term welfare benefits and improved management efficiency achieved through structured rehabilitation. Sharing technical resources and training programs among institutions would help reduce costs and promote wider adoption of welfare-centered rehabilitation frameworks.

## Conclusions

This study evaluated the outcomes of a structured rehabilitation program for a tail-fluke-amputated Indo-Pacific bottlenose dolphin through a structured, multi-phase program aimed at restoring swimming ability, social integration, and psychological well-being. We hypothesized that a welfare-centered intervention combining prosthetic-assisted locomotor training and socially enriching environments could lead to long-term functional and behavioral recovery. Our findings support this hypothesis.

The use of custom-made prosthetic tail flukes significantly improved propulsion per stroke (RPDS), even though absolute swim speed remained suboptimal due to mechanical constraints. Notably, vertical tail-beat locomotion was retained even after prosthesis discontinuation, indicating durable motor adaptation. Socially, affiliative behavior increased to levels surpassing those observed in healthy conspecifics, with no aggression or abnormal behaviors recorded, suggesting a complete reintegration into the social group. These improvements were sustained until the final stage of life, highlighting the lasting impact of rehabilitation.

This study contributes new insights to the field of marine mammal welfare by (1) providing quantitative evidence of long-term biomechanical and social adaptation following tail loss, (2) demonstrating the value of prosthetic training beyond physical function, including psychological resilience and enhanced self-efficacy, and (3) proposing a framework that integrates physical, social, and psychological metrics for evaluating rehabilitation success.

Limitations of this study include its single-subject design and the inability to assess long-term prosthetic use due to medical decisions aimed at preventing the progression of infection. While these constraints are inherent to case-based studies, they also highlight the need for broader, collaborative efforts to validate and extend the present findings. To overcome the limitations of single-case studies, future research should involve multi-institutional collaborations that allow comparative and longitudinal assessments of prosthetic adaptation and social reintegration. Establishing shared behavioral metrics and rehabilitation records across facilities would enable the identification of individual variability and key factors for success, contributing to a more generalizable welfare assessment framework. Such collaborative efforts will strengthen

the scientific basis of marine mammal rehabilitation and promote higher international standards of welfare practice. It is also worth speculating that improved motor patterns may have facilitated social acceptance by reducing visual incongruence—a hypothesis that warrants further ethological investigation.

Taken together, this study underscores the feasibility and necessity of holistic rehabilitation approaches for cetaceans with severe physical trauma. By addressing mobility, sociality, and emotional health in tandem, such programs can meaningfully improve quality of life and serve as a model for welfare-driven innovation in marine animal care.

## Supporting information

**S1 Video. This supplementary video illustrates the structure and attachment procedure of the artificial tail fluke used during Phase 1 of rehabilitation.** The prosthesis consisted of a rubber tail fluke connected to a carbon fiber socket via a metal plate. The socket was divided into upper and lower sections, which were joined and secured with bolts. A silicone lining was inserted inside the socket to prevent abrasion of the caudal peduncle and to improve comfort and safety during attachment.
(MP4)

**S2 Video. Video recording of the subject dolphin exhibiting pronounced lateral body undulation following tail fluke loss.** Although a prosthetic tail is attached, the vertical tail-beat motion had not yet been adequately rehabilitated. As a result, the observed swimming behavior deviates from the typical pattern and imposes increased biomechanical strain on the axial body musculature.
(MP4)

**S3 Video. Video footage depicting the rehabilitation procedure aimed at restoring vertical tail-beat motion.** As described in the main text, the dolphin's body is gently restrained while a trainer manually guides the distal tail region through its full vertical range of motion to promote flexibility and reestablish proper stroke axis.
(MP4)

**S4 Video. Video recording of the subject dolphin exhibiting stable and coordinated swimming while wearing the prosthetic tail flukes.** The footage illustrates successful habituation to the device, achieved during Rehabilitation Phase 2.
(MP4)

**S5 Video. Video footage of the subject dolphin engaged in affiliative social behaviors after Rehabilitation Phase 2.** The clip shows consistent synchronous swimming and close-contact positioning with conspecifics, as well as an instance of drafting behavior—defined as swimming in the hydrodynamic wake of another dolphin. These behaviors reflect the reestablishment of social cohesion and affiliative engagement following rehabilitation.
(MP4)

**S6 Table. The raw data used to generate the figures and statistical results in the manuscript are provided as a supplementary file.** This includes behavioral observations and swimming performance measurements necessary to replicate the analyses.
(XLSX)

## Acknowledgments

We thank the staff at the Okinawa Churaumi Aquarium for animal training, handling, and medical treatment. Koji Tokutake and Dr. Isao Kawazu, former managers of the marine mammal department, provided encouragement and support throughout this project. We also thank Bridgestone Corporation for their collaboration in the development of the prosthetic

tail. We are also grateful to the staff of the Wildlife Research Center of Kyoto University for kindly providing the data loggers used to measure swimming performance. We also express our sincere respect to Sami, the Indo-Pacific Bottlenose Dolphin who participated in this study. Although she passed away due to a lung disease during the manuscript preparation, her contributions to our understanding of physical and social recovery in marine mammals remain invaluable. Artificial intelligence assistance (ChatGPT, OpenAI) was used solely for language polishing and improving the clarity of the English text. All scientific content, data analysis, and conclusions were developed entirely by the authors.

## Author contributions

**Conceptualization:** Suguru Higa, Keiichi Ueda, Yukinori Nakakita, Shin-ichiro Oka.

**Data curation:** Suguru Higa, Sayaka Takahashi, Eri Nakashima, Haruka Ikeshima, Nihiro Adachi, Shin-ichiro Oka.

**Formal analysis:** Nihiro Adachi, Shin-ichiro Oka.

**Investigation:** Suguru Higa, Sayaka Takahashi, Eri Nakashima, Yui Kurosu, Haruka Ikeshima, Ryota Yagi, Hirobumi Umeyama, Hitoshi Yamamoto, Yukinori Nakakita, Kazuma Tochigi, Taihei Kagawa.

**Project administration:** Suguru Higa, Keiichi Ueda, Hirobumi Umeyama, Shin-ichiro Oka.

**Supervision:** Keiichi Ueda, Shin-ichiro Oka.

**Visualization:** Shin-ichiro Oka.

**Writing – original draft:** Shin-ichiro Oka.

**Writing – review & editing:** Sayaka Takahashi, Eri Nakashima, Yui Kurosu, Haruka Ikeshima, Ryota Yagi, Nihiro Adachi, Keiichi Ueda, Hirobumi Umeyama, Hitoshi Yamamoto, Yukinori Nakakita, Kazuma Tochigi, Taihei Kagawa, Shin-ichiro Oka.

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
