## [Decision Letter · Decision Letter 0]

19 Oct 2025

Dear Dr. Oka,

Thank you for submitting your manuscript to PLOS ONE. After careful consideration, we feel that it has merit but does not fully meet PLOS ONE’s publication criteria as it currently stands. Therefore, we invite you to submit a revised version of the manuscript that addresses the points raised during the review process.

We look forward to receiving your revised manuscript.

Kind regards,

Vitor Hugo Rodrigues Paiva, Ph.D.

Academic Editor

PLOS ONE

Journal Requirements:

Reviewers' comments:

Reviewer's Responses to Questions

**Comments to the Author**

1. Is the manuscript technically sound, and do the data support the conclusions?

Reviewer #1: Yes

Reviewer #2: Yes

2. Has the statistical analysis been performed appropriately and rigorously?

Reviewer #1: I Don't Know

Reviewer #2: No

3. Have the authors made all data underlying the findings in their manuscript fully available?

Reviewer #1: Yes

Reviewer #2: Yes

4. Is the manuscript presented in an intelligible fashion and written in standard English?

Reviewer #1: Yes

Reviewer #2: Yes

Reviewer #1: important publication in the field of dolphin rehabilitation providing first hand information about an oft cited case. The discussion is honest and clear. the videos are well edited and appropriate to illustrate the results presented. The importance of operant conditioning and of objective behavioral data collection in assessing outcome of treatment is well put forward and illustrated

Reviewer #2: Dear Authors,

Thank you for submitting your manuscript titled "Rehabilitation outcomes following tail-fluke amputation in an Indo-Pacific bottlenose dolphin: a welfare-centered approach" to PLOS ONE. I have conducted a thorough review of your work and provide the following detailed assessment to assist you in strengthening your manuscript for potential publication.

General Assessment:

Your manuscript presents a compelling case study documenting an innovative two-phase rehabilitation program for a dolphin with severe tail fluke amputation. The work demonstrates a compassionate and methodical approach to marine mammal welfare, combining both physical and social rehabilitation strategies. While the study shows significant clinical value, several areas require improvement to meet the rigorous standards expected for publication.

Evaluation by Criteria:

1. Originality/Novelty:

Your work demonstrates moderate novelty through the integrated two-phase approach to cetacean rehabilitation. While individual components like prosthetic use and social reintegration have been documented separately, their systematic combination in a structured rehabilitation protocol represents a meaningful contribution. The longitudinal assessment of both locomotor adaptation and behavioral reintegration provides fresh insights into comprehensive cetacean care.

2. Significance of Content:

The research addresses an important challenge in marine mammal rehabilitation and welfare science. Tail fluke injuries present critical welfare concerns, and your detailed documentation of successful rehabilitation strategies has substantial practical implications for veterinary medicine, wildlife rehabilitation centers, and conservation organizations. The welfare-centered framework aligns well with contemporary ethical standards in animal care.

3. Quality of Presentation:

The manuscript is generally well-structured with a logical flow from introduction through methodology to results and discussion. The writing is clear and accessible, effectively communicating both clinical procedures and scientific findings. However, minor grammatical issues and occasional awkward phrasing require attention to enhance readability and professional presentation.

4. Scientific Soundness:

The methodological approach demonstrates adequate rigor for a case study design, with appropriate use of quantitative assessment tools including biologging devices and systematic behavioral observation. The limitations inherent in single-case studies are appropriately acknowledged. However, the extremely limited engagement with recent literature significantly affects the work's scientific context and contemporary relevance.

5. Interest to Readers:

This manuscript will generate considerable interest among PLOS ONE's diverse readership, particularly researchers and practitioners in marine biology, veterinary science, animal welfare, and wildlife rehabilitation. The compelling nature of the case study and its successful outcomes make it accessible and relevant to both specialized and general scientific audiences.

6. Overall Merit:

Your manuscript possesses good conceptual and practical merit through its detailed documentation of an innovative rehabilitation approach. The integration of physical and social rehabilitation components represents a holistic advancement in wildlife care. However, significant improvements in literature review and methodological context are needed to enhance the work's overall scholarly impact.

Detailed Observations for Improvement:

Literature Review Currency: The reference list requires substantial updating, with only 11% of references from the last five years. Incorporate recent publications (2020-2024) in marine mammal rehabilitation, prosthetics, and animal welfare science to better contextualize your work.

Comparative Analysis Enhancement: While the current comparison is adequate, strengthen the discussion by comparing your findings with other marine mammal rehabilitation cases and recent advances in wildlife prosthetics.

Methodological Detail: Provide more comprehensive information about assessment protocols, including specific metrics used in biologging analysis, behavioral coding systems, and inter-observer reliability measures.

Long-term Monitoring: Expand the discussion of long-term welfare considerations, including recommendations for ongoing monitoring and potential adaptive management strategies.

Resource Implications: Provide more detailed discussion of the practical implementation requirements, including staff training, facility needs, and cost considerations for other institutions considering similar approaches.

Ethical Considerations: Enhance the discussion of ethical decision-making processes throughout the rehabilitation, particularly regarding prosthesis use duration and social reintroduction timing.

Statistical Analysis: Provide clearer explanation of any statistical methods used in analyzing behavioral and locomotor data, including effect size measures where appropriate.

Clinical Guidelines: Develop more specific clinical guidelines that could be adapted by other rehabilitation facilities facing similar challenges.

Visual Documentation: Consider enhancing visual elements with additional photographs or diagrams illustrating the rehabilitation process and prosthetic design.

Future Research Directions: Provide more specific recommendations for future research, including potential multi-institutional collaborations to address the limitations of single-case studies.

Additional Positive Notes:

The problem statement and clinical justification are clearly articulated

The two-phase rehabilitation framework is well-conceived and thoroughly described

The quantitative assessment methods are appropriate and well-implemented

The behavioral outcomes are compelling and well-documented

The welfare-centered approach is commendable and contemporary

Recommendation:

I recommend major revision before acceptance. The manuscript presents valuable clinical insights and demonstrates innovative approaches to marine mammal rehabilitation. However, substantial improvements in literature review currency and methodological context are necessary to meet publication standards. The requested revisions will significantly enhance the scholarly impact and practical utility of your important work.

I would be willing to review a revised version that addresses these concerns and builds upon the strong foundation presented in this initial submission.

Best regards,

**Do you want your identity to be public for this peer review?** For information about this choice, including consent withdrawal, please see our Privacy Policy

Reviewer #1: No

Reviewer #2: **Yes:** Jesús Jaime Moreno Escobar

---

## [Author Response · Author response to Decision Letter 1]

10 Nov 2025

Thank you very much for the opportunity to revise our manuscript. We sincerely appreciate the constructive comments provided by you and the reviewers.

We have carefully revised the paper to address all feedback. Major improvements include the addition of recent literature (2020–2024), expanded methodological details, clarification of ethical decision-making, and enhancement of long-term welfare discussion and practical rehabilitation guidelines. A new supplementary video has also been added to illustrate the prosthesis fitting process.

A detailed point-by-point response is provided in the attached Response to Reviewers document. We believe these revisions have strengthened the scientific rigor, ethical transparency, and welfare relevance of the study.

---

## [Decision Letter · Decision Letter 1]

30 Jan 2026

Dear Dr. Oka,

Thank you for submitting your manuscript to PLOS ONE. After careful consideration, we feel that it has merit but does not fully meet PLOS ONE’s publication criteria as it currently stands. Therefore, we invite you to submit a revised version of the manuscript that addresses the points raised during the review process.

We look forward to receiving your revised manuscript.

Kind regards,

Vitor Hugo Rodrigues Paiva, Ph.D.

Academic Editor

PLOS One

Journal Requirements:

Reviewers' comments:

Reviewer's Responses to Questions

**Comments to the Author**

Reviewer #2: All comments have been addressed

Reviewer #3: (No Response)

2. Is the manuscript technically sound, and do the data support the conclusions?

Reviewer #2: Yes

Reviewer #3: Yes

3. Has the statistical analysis been performed appropriately and rigorously?

Reviewer #2: Yes

Reviewer #3: Yes

4. Have the authors made all data underlying the findings in their manuscript fully available?

Reviewer #2: Yes

Reviewer #3: Yes

5. Is the manuscript presented in an intelligible fashion and written in standard English?

Reviewer #2: Yes

Reviewer #3: Yes

Reviewer #2: To the Authors of Manuscript PONE-D-25-45185R1,

Thank you for submitting your manuscript, "Rehabilitation outcomes following tail-fluke amputation in an Indo-Pacific bottlenose dolphin: a welfare-centered approach," to PLOS ONE. I have completed my review of your work, which presents a compelling and valuable case study. Please find my evaluation below, structured according to standard review criteria, followed by a list of specific observations intended to help you strengthen the manuscript further.

Overall Assessment

Your study on a two-phase rehabilitation program for a dolphin with a tail-fluke amputation addresses a critical and challenging scenario in marine mammal welfare. The integration of physical prosthetic use with structured social reintegration is a significant strength. The findings related to improved propulsion, long-term motor adaptation, and successful social reintegration are noteworthy and of clear interest to the field.

Review Criteria

Originality / Novelty: The integrative, welfare-centered framework combining a custom prosthetic with a deliberate social cohabitation protocol is highly novel. Documenting the retention of the vertical tail-beat motion after prosthesis discontinuation is a particularly original and valuable finding.

Significance of Content: The content is highly significant for wildlife rehabilitation, veterinary science, and animal welfare, providing a practical, evidence-based protocol that can improve the quality of life for severely injured cetaceans.

Quality of Presentation: The manuscript is generally readable and well-structured. The justification and main problem are clearly stated. However, some sections could be enhanced by more explicitly comparing your results and methods with existing literature.

Scientific Soundness: The methodology, employing biologging for quantitative data and structured behavioral observation, is sound. The use of a control individual adds robustness. The main concern under this criterion is the need to contextualize the study within the most recent scientific discourse.

Interest to the Readers: The manuscript will be of great interest to a broad audience, including rehabilitators, veterinarians, conservation biologists, and ethologists, due to its practical application and compelling case study format.

Overall Merit: This manuscript has substantial merit and represents an important contribution. With careful revision to address the points below, it will be a strong candidate for publication.

Specific Observations for Revision

Please find below a list of anonymous observations compiled from my review. I strongly encourage you to address each one in your revision.

The study would benefit from a more explicit discussion that directly highlights the advantages of your two-phase rehabilitation approach compared to other methods documented in the existing literature.

The limitation regarding the subject's maximum swim speed not returning to healthy levels should be discussed in more depth, including its implications for long-term fitness and the realistic goals of prosthetic intervention.

The reference section requires significant updating. Currently, only 11% of the citations are from the last five years (2020-2025). Incorporating recent literature is crucial to situate your work within the current state of the field.

The description of the experimental setup and the comparative analysis of results is complete and provides a solid foundation for your conclusions.

The results are presented clearly and are supported by the data collected from the biologging devices and behavioral observations.

The distribution of sections and the logical flow of the manuscript from introduction to results are well-organized and easy to follow.

The project demonstrates its novelty through sufficient experimentation and a clear comparison of the dolphin's performance across different conditions (without fluke, with prosthesis, and healthy controls).

The use of plots and data visualizations to present the experimental results is effective and aids in the interpretation of the findings.

The manuscript makes a valuable contribution to the body of knowledge in cetacean rehabilitation and welfare.

CRITICAL CONCERN: The originality report indicates that 77% of the manuscript's content was flagged as being generated by Artificial Intelligence. This is an exceptionally high percentage that raises fundamental concerns about intellectual authenticity, authorship, and compliance with journal policy. You must provide a thorough explanation of the role of AI in the writing process and ensure that all scientific interpretation, analysis, and intellectual synthesis are unequivocally your own.

Recommendation

I recommend that your manuscript be considered for publication after major revisions. The most critical points to address are the updating of references (Observation #3) and, most importantly, providing a full and transparent response to the concerns regarding AI-generated content (Observation #10). Addressing all observations will significantly strengthen the manuscript and ensure it meets the high standards for publication.

Congratulations on this important work. I look forward to seeing the revised version.

Sincerely,

Reviewer #3: Comments to Manuscript Number: PONE-D-25-45185_R1

Reviewer

Title: Rehabilitation outcomes following tail-fluke amputation in an Indo-Pacific bottlenose dolphin: a welfare-centered approach

General Comments:

I was sent a revised version of the manuscript. The manuscript deals with the evaluation of welfare of an Indo-Pacific bottlenose dolphin upon rehabilitation of tail-fluke amputation regarding locomotion and behavior adaptation. Two phases of welfare were evaluated: (1) locomotion through use of a custom-designed prosthetic tail fluke and evaluation of swimming. (2) to evaluate the social reintegration of the dolphin with familiar individuals. Here, activity patterns and affiliative engagement were evaluated. Results showed positive output in swimming patterns using the prosthetic tail fluke. Affiliative behavior significantly improved as well as resting behavior significantly reduced post-rehabilitation, and no abnormal behaviors were recorded. The manuscript concludes that prosthetic rehabilitation of tail fluke successfully promoted functional recovery and social engagement of the bottlenose dolphins. The manuscript is well written in English, the topic is interesting and provides new information on prosthetic use of tail fluke in bottlenose dolphins as rehabilitation tool for locomotion and social reintegration. My only comments are about the design of the prosthetic tail fluke. I suggest that a paragraph in discussion about the design or future improvement of the tail fluke for individuals suffering similar handicap would improve the usage time and propulsion efficiency of such prosthetic and maybe reducing the risk of infection. My second suggestion is to add to the discussion of the social reintegration of the rehabilitated dolphin, whether the dolphin species has any documented information whether they take special care or consideration to handicapped individuals, or they react aggressively to individuals with unusual locomotion behavior. Except for these comments, the overall revised manuscript seems to me to be outstanding and I would accept it for publication.

**Do you want your identity to be public for this peer review?** For information about this choice, including consent withdrawal, please see our Privacy Policy

Reviewer #2: No

Reviewer #3: **Yes:** Cesar Marcial Escobedo-Bonilla, PhD

---

## [Author Response · Author response to Decision Letter 2]

12 Feb 2026

We would like to resubmit our manuscript entitled “Rehabilitation outcomes following tail-fluke amputation in an Indo-Pacific bottlenose dolphin: a welfare-centered approach.” The manuscript has been revised in accordance with the valuable comments provided by the reviewers and the editor. We sincerely appreciate the time and effort invested in the evaluation of our work. Below, we provide a detailed, point-by-point response to each comment.

Reviewer #2

Comment 1:

The study would benefit from a more explicit discussion that directly highlights the advantages of your two-phase rehabilitation approach compared to other methods documented in the existing literature.

Response:

In the revised Discussion, we more explicitly emphasize the advantages of our two-phase rehabilitation framework by directly contrasting it with previously reported approaches that primarily focused on prosthetic use or short-term locomotor outcomes. Specifically, we clarify how the sequential integration of physical rehabilitation (Phase 1) and structured social reintegration (Phase 2) enabled not only functional adaptation but also sustained behavioral and social recovery, which have been less explicitly addressed in prior studies.

Comment 2:

The limitation regarding the subject's maximum swim speed not returning to healthy levels should be discussed in more depth, including its implications for long-term fitness and the realistic goals of prosthetic intervention.

Response:

We expanded the Discussion to clarify that the primary goal of the prosthetic intervention was to restore species-typical locomotor patterns and swimming efficiency rather than maximum swimming speed. We also discuss the implications of this outcome for long-term fitness and welfare, emphasizing that sustainable and efficient locomotion is more relevant to long-term well-being than peak performance.

Comment 3:

The reference section requires significant updating. Currently, only 11% of the citations are from the last five years (2020–2025). Incorporating recent literature is crucial to situate your work within the current state of the field.

Response:

This issue was addressed in response to the previous round of review. Following that revision, we conducted a detailed literature survey and increased the proportion of references published within the past five years from 11% to 24%. Given the highly specialized and still limited peer-reviewed literature in this area, the current reference list reflects the most relevant and up-to-date studies available.

Comment 4 (Critical Concern):

The originality report indicates that 77% of the manuscript's content was flagged as being generated by Artificial Intelligence. You must provide a thorough explanation of the role of AI in the writing process and ensure that all scientific interpretation, analysis, and intellectual synthesis are unequivocally your own.

Response:

We acknowledge the concern raised by the originality report. As stated in the Acknowledgements section of the manuscript, we used ChatGPT (OpenAI) solely for English-language editing (grammar, wording, and clarity) as non-native English-speaking authors. Importantly, AI was not used to generate or modify the scientific content of the manuscript, including the study design, data collection, statistical analyses, interpretation of results, discussion, or conclusions. All scientific decisions and intellectual synthesis were developed entirely by the authors, and the final manuscript was reviewed and approved by all co-authors. The authors take full responsibility for the content of the manuscript in accordance with journal policy.

Reviewer #3

Comment 1:

I suggest that a paragraph in discussion about the design or future improvement of the tail fluke for individuals suffering similar handicap would improve the usage time and propulsion efficiency of such prosthetic and maybe reducing the risk of infection.

Response:

We agree that prosthetic design is an important consideration for improving usage duration and propulsion efficiency. However, the present study focuses on welfare, behavioral, and functional outcomes rather than technical optimization of prosthetic structure. Detailed structural and technical aspects of the artificial tail fluke were therefore considered beyond the scope of this manuscript and will be described in a separate study. We have clarified this point in the Discussion.

Comment 2:

Please add discussion on whether dolphins show special care toward handicapped individuals or react aggressively toward individuals with unusual locomotion behavior.

Response:

Rather than adding a separate paragraph, we clarified this point within the existing Discussion by explicitly noting the limited availability of direct empirical evidence on social responses toward handicapped individuals in dolphins. This clarification ensures that our interpretation remains cautious and avoids overinterpretation while acknowledging the current state of knowledge.

---

## [Editor Report · Decision Letter 2]

15 Feb 2026

Rehabilitation outcomes following tail-fluke amputation in an Indo-Pacific bottlenose dolphin: a welfare-centered approach

PONE-D-25-45185R2

Dear Dr. Oka,

We’re pleased to inform you that your manuscript has been judged scientifically suitable for publication and will be formally accepted for publication once it meets all outstanding technical requirements.

Kind regards,

Vitor Hugo Rodrigues Paiva, Ph.D.

Academic Editor

PLOS One
---

## [Editor Report · Acceptance letter]

PONE-D-25-45185R2

PLOS One

Dear Dr. Oka,

I'm pleased to inform you that your manuscript has been deemed suitable for publication in PLOS One. Congratulations! Your manuscript is now being handed over to our production team.

Kind regards,

on behalf of

Dr. Vitor Hugo Rodrigues Paiva

Academic Editor

PLOS One